# A Finite Element Method for Modeling Diffusion and Drug Release from Nanocellulose/Nanoporous Silicon Composites

**DOI:** 10.3390/pharmaceutics17010120

**Published:** 2025-01-16

**Authors:** Paulo Zúñiga, Marcelo Aravena, Silvia Ponce, Jacobo Hernandez-Montelongo

**Affiliations:** 1Department of Mathematical and Physical Sciences, Catholic University of Temuco, Temuco 4813302, Chile; maravena2021@alu.uct.cl; 2Institute of Scientific Research IDIC, University of Lima, Lima 15023, Peru; sponce@ulima.edu.pe; 3Department of Chemical Engineering, University of Guadalajara, Guadalajara 44430, Mexico

**Keywords:** finite element method, drug delivery, composites, nanocellulose, nanoporous silicon

## Abstract

**Background and Objective**: A previous study investigated the in vitro release of methylene blue (MB), a widely used cationic dye in biomedical applications, from nanocellulose/nanoporous silicon (NC/nPSi) composites under conditions simulating body fluids. The results showed that MB release rates varied significantly with the nPSi concentration in the composite, highlighting its potential for controlled drug delivery. To further analyze the relationship between diffusion dynamics and the MB concentration, this study developed a finite element (FE) method to solve Fick’s equations governing the drug delivery system. **Methods**: Release profiles of MB from NC/nPSi composites with varying nPSi concentrations (0%, 0.1%, 0.5%, and 1.0%) were experimentally measured in triplicate using phosphate-buffered saline (PBS) at 37 °C, pH 7.4, and 100 rpm. Mathematical models incorporating linear and quadratic dependencies of the diffusion coefficient on the MB concentration were developed and tested using the FE method. Model parameters were refined by minimizing the error between simulated and experimental MB release profiles. **Results**: The proposed FE method closely matched experimental data, validating its accuracy and robustness in simulating the diffusion and release processes. **Conclusions**: This study emphasizes the significant impact of the nPSi concentration on enhancing release control and highlights the importance of material composition in designing drug delivery systems. The findings suggest that the FE method can be effectively applied to model other complex systems, paving the way for advancements in precision drug delivery and broader biomedical applications.

## 1. Introduction

Nanoporous silicon (nPSi) stands out as an excellent biomaterial for drug delivery applications due to its high surface area, biocompatibility, biodegradability, and bioresorbability [1,2,3]. Typically, drugs are either loaded into the porous matrix or immobilized on the surface following appropriate surface derivatization. When combined with biopolymers, it acts as a substrate for composite materials, introducing advantageous chemical and physical properties not present in individual components. These benefits include improved control over drug release kinetics and enhanced stability in aqueous solutions [4,5]. As a result, nPSi has been paired with various organic matrices to create composites as advanced drug delivery systems, such as β-cyclodextrin polymers [4,6,7], oxidized hyaluronic acid hydrogels [8], and poly(L-lactide) acid [9], among others.

In this context, nanocellulose (NC) has recently emerged as one of the most promising “green” materials for obtaining drug delivery carriers as composites. It offers adaptable surface chemistry, a high surface area, biocompatibility, and biodegradability [10,11]. Recently, K. Garrido-Miranda et al. [12] synthesized NC/nPSi composites for the controlled release of methylene blue (MB), a cationic thiazine dye widely used in biomedicine for various purposes. These include the treatment of methemoglobinemia [13], its use as a marker and indicator in various surgical techniques [14], and its use as an analgesic in different treatments [15]. Additionally, it has been applied as an antibacterial, antiviral agent, and against cancer cells [16].

On the other hand, the finite element (FE) method is a numerical technique commonly used to approximate the solution of ordinary and partial differential equations (see, e.g., [17,18,19] and the references therein). It involves writing the equation in its variational (weak) form and approximating the solution, originally defined in an infinite-dimensional space, by restricting it to a finite-dimensional subspace. This approach has been successfully applied to simulate various physical phenomena, including the fluid–structure interaction [20], electromagnetism [21], and heat transfer [22].

This work presents an FE method to simulate the diffusion and controlled release of MB from NC/nPSi composites. The simulations are based on Fick’s second law [23], a widely used model for release kinetics (e.g., [24,25,26,27,28]). In this study, a concentration-dependent diffusion coefficient is incorporated to more accurately represent the behavior of highly polymerized NC. Furthermore, the influence of temperature on diffusivity in biological environments is considered a critical factor. This temperature dependency is modeled using the Stokes–Einstein relationship [29], offering a deeper insight into the dynamics of the diffusion process:
$$\begin{array}{c} D=\frac{k_{B}T}{a\eta r}, \end{array}$$
where $k_{B}$ is Boltzmann’s constant, *T* represents the temperature, *a* denotes the viscosity, η is a constant that accounts for the boundary conditions between the diffusing molecule and the solvent, and *r* is the radius of the molecule. Consequently, diffusion coefficients increase with temperature, as observed in drug delivery experiments [30,31].

Nevertheless, as this model is further refined by incorporating dependency relations from [32], which align with the experimental release profiles reported by K. Garrido-Miranda et al. [12], the influence of temperature was not investigated and remains limited to a concentration-dependent diffusion coefficient.

Fick’s second law is first discretized using the FE method, transforming the governing equation into a discrete system. Since the diffusion coefficient depends on concentration, the non-linearities are addressed using the Picard iteration. This method linearizes the system at each time step, enabling its resolution using Octave [33]. The proof of unique solvability of the system is provided under the assumption of a bounded diffusion coefficient. Based on this assumption, examples are presented to offer valuable insights into designing more efficient drug delivery systems for biomedical applications.

The outline of this article is as follows. Section 2 presents the model problem and its FE discretization. A case study comparing the numerical solution with experimental data is presented in Section 3. Numerical details concerning future directions are discussed in Section 4, while conclusions are drawn in Section 5.

## 2. Materials and Methods

### 2.1. Model Problem

This work is about the one-dimensional diffusion and controlled release of MB through an NC/nPSi composite of thickness 2L (see Figure 1). It is assumed that *L* is sufficiently small, such that the amount of MB passing through the edges of the composite is negligible.

In this context, the drug delivery system is modeled within the framework of Fick’s second law [23], a widely used approach in diffusion processes, with a diffusion coefficient *D* that depends on the concentration ϕ. Let I=(−L,L) be an interval along the *x*-axis, and let T>0 be a given finite time. The governing equation is(1)$$\frac{\partial \phi }{\partial t}=\frac{\partial }{\partial x}\left(D\left(\phi \right)\frac{\partial \phi }{\partial x}\right)\;\mathrm{in}\;I\times \left(0,T\right).$$

At the surfaces x=±L, the concentration is prescribed as $\phi =\phi_{\infty }$ for all t∈(0,T).

When the concentration is initially uniform, with $\phi \left(x,0\right)=\phi_{0}$ for all x∈I, and both *D* and $\phi_{\infty }$ are constants, the partial differential Equation (Equation 1) can be solved using the method of separation of variables or the Laplace transform, as detailed in [34]. This gives(2)$$\frac{\phi \left(x,t\right)-\phi_{0}}{\phi_{\infty }-\phi_{0}}=1-\frac{4}{\pi }\sum_{n=0}^{\infty }\frac{{\left(-1\right)}^{n}}{\left(2n+1\right)}\cos[\frac{(2n+1)\pi x}{2L}]\exp[-\frac{D{\left(2n+1\right)}^{2}\pi^{2}t}{4L^{2}}].$$

Moreover, if $M_{t}$ denotes the amount of MB released at time *t* and $M_{\infty }$ the corresponding quantity after infinite time, the release profile is given by(3)$$\frac{M_{t}}{M_{\infty }}=1-\frac{8}{\pi^{2}}\sum_{n=0}^{\infty }\frac{1}{{\left(2n+1\right)}^{2}}\exp[-\frac{D{\left(2n+1\right)}^{2}\pi^{2}t}{4L^{2}}].$$

Although assuming a constant *D* is common in diffusion modeling, this assumption is unrealistic for highly polymerized substances [34], such as NC and NC-based composites. Given the relevance of the NC in this context, Fick’s second law in (Equation 1) with a variable *D* will be addressed in the following sections, focusing on the numerical solution. To facilitate understanding, we first analyze the case with a constant *D* and then extend the analysis to accommodate D=D(ϕ).

### 2.2. Constant Diffusion Coefficient

#### 2.2.1. Preliminaries

Let us review some basic concepts of functional analysis which are useful in dealing with partial differential equations. We first define the Sobolev space $H^{1}\left(I\right)$ as$$H^{1}\left(I\right):=\left\{\psi \in L^{2}\left(I\right):\;\psi^{\prime }\in L^{2}\left(I\right)\right\},$$
which is a Hilbert space equipped with the inner product$${\left(\phi ,\psi \right)}_{1,I}:=\int_{I}\phi \psi dx+\int_{I}\phi^{\prime }\psi^{\prime }dx\;\forall \;\phi ,\psi \in H^{1}\left(I\right).$$

For further details, we refer the reader to [35]. The norm induced by ${\left(\;\cdot \;,\;\cdot \;\right)}_{1,I}$ is given by$${∥\psi ∥}_{1,I}:={\left({∥\psi ∥}_{0,I}^{2}+{\left|\psi \right|}_{1,I}^{2}\right)}^{1/2}\;\forall \;\psi \in H^{1}\left(I\right),$$
where ${\left|\psi \right|}_{1,I}:={∥\psi^{\prime }∥}_{0,I}$ is a semi-norm on $H^{1}\left(I\right)$. It is well known that the quantity ${\left|\psi \right|}_{1,I}$ is a norm equivalent to ${∥\psi ∥}_{1,I}$ on the following subspace of $H^{1}\left(I\right)$:(4)$$H_{0}^{1}\left(I\right):=\left\{\psi \in H^{1}\left(I\right):\;\psi =0\;\mathrm{at}\;x=\pm L\right\},$$
due to Poincaré’s inequality. This result is stated below, and its proof can be found in ([35], Proposition 8.13).

**Lemma** **1**(Poincaré’s inequality). *Let I be a bounded interval. Then, there exists a constant $C_{P}>0$, depending only on I, such that*(5)$${∥\psi ∥}_{1,I}\leq C_{P}{\left|\psi \right|}_{1,I}\;\forall \;\psi \in H_{0}^{1}\left(I\right).$$

Finally, we note that an alternative way of interpreting ϕ in (Equation 1) is to treat it as a function of time, taking values in a Sobolev space, e.g., *V*, where the elements of *V* are functions that depend only on the spatial variable:ϕ:t∈(0,T)↦ϕ(t)≡ϕ(·,t)∈V.

This notation will be used throughout this work. Furthermore, time derivatives will be denoted by $d_{t}\left(\;\cdot \;\right)$.

#### 2.2.2. Weak Formulation

To introduce the weak formulation of Fick’s second law, we multiply (Equation 1) by a test function $\psi \in H_{0}^{1}\left(I\right)$, assume a constant *D*, and perform integration over *I*, yielding
$$\int_{-L}^{L}\psi d_{t}\phi \left(t\right)dx+D\int_{-L}^{L}\phi^{\prime }\left(t\right)\psi^{\prime }dx-{D\phi^{\prime }\left(t\right)\psi |}_{x=-L}^{x=L}=0.$$

Then, using the condition that ψ=0 at x=±L (cf. (Equation 4)), we obtain(6)$$\int_{-L}^{L}\psi d_{t}\phi \left(t\right)dx+D\int_{-L}^{L}\phi^{\prime }\left(t\right)\psi^{\prime }dx=0.$$

By virtue of identity (Equation 6) and under the conditions used to determine the exact ϕ in (Equation 2), the weak formulation of Equation (Equation 1) with a constant *D* reads the following: For almost every t∈(0,T), find $\phi \left(t\right)\in H^{1}\left(I\right)$, such that $\phi =\phi_{\infty }$ on {−L,L}×(0,T), $\phi =\phi_{0}$ on I×{0}, and (7)$$\int_{-L}^{L}\psi d_{t}\phi \left(t\right)dx+D\int_{-L}^{L}\phi^{\prime }\left(t\right)\psi^{\prime }dx=0\;\forall \;\psi \in H_{0}^{1}\left(I\right).$$

At this point, we emphasize that the FE method, which will be detailed later, does not directly allow us to deduce the unique solvability for the discretization of (Equation 7) when $\phi_{\infty }\neq 0$. This is because, in this case, the discrete versions of ϕ(t) and ψ reside in different spaces. To address this issue, we recall that the orthogonal complement of $H_{0}^{1}\left(I\right)$ in $H^{1}\left(I\right)$ is defined by(8)$$H_{0}^{1}{\left(I\right)}^{⊥}:=\{\lambda \in H^{1}\left(I\right):\;{\left(\lambda ,\psi \right)}_{1,I}=0\;\forall \;\psi \in H_{0}^{1}\left(I\right)\}.$$

It follows that $\lambda \in H_{0}^{1}{\left(I\right)}^{⊥}$ if and only if λ is the weak solution to the equation$$-\lambda^{\prime \prime }+\lambda =0.$$

Accordingly, we decompose $H^{1}\left(I\right)$ as the direct sum $H_{0}^{1}\left(I\right)⊕W$, where *W* is the subspace spanned by $\left\{e^{x},e^{-x}\right\}$. Next, we introduce the auxiliary unknown(9)ϑ(t):=ϕ(t)−λ,
with $\lambda \in H_{0}^{1}{\left(I\right)}^{⊥}$ being defined by(10)$$\lambda \left(x\right):=(\frac{\phi_{\infty }}{e^{L}+e^{-L}})\left(e^{x}+e^{-x}\right).$$

It is easy to check that $\lambda \left(-L\right)=\lambda \left(L\right)=\phi_{\infty }$, from which we deduce that (Equation 7) is equivalent to the following problem: for almost every t∈(0,T), find $\vartheta \left(t\right)\in H_{0}^{1}\left(I\right)$ such that $\vartheta =\phi_{0}$ on I×{0} and(11)$$\int_{-L}^{L}\psi d_{t}\vartheta \left(t\right)dx+D\int_{-L}^{L}\vartheta^{\prime }\left(t\right)\psi^{\prime }dx=-D\int_{-L}^{L}\lambda^{\prime }\psi^{\prime }dx\;\forall \;\psi \in H_{0}^{1}\left(I\right).$$

We can therefore use ϑ to recover the solution to (Equation 7). In particular, ϑ=ϕ when $\phi_{\infty }=0$.

### 2.3. Discretization of the Model with a Constant Diffusion Coefficient

This section focuses on approximating the solution to (Equation 11) using a fully discrete scheme that combines an FE method in space with an implicit Euler method in time. For simplicity, homogeneous boundary conditions are initially assumed. The non-homogeneous case will be addressed in Section 2.3.3.

#### 2.3.1. Fully Discrete Scheme

Let $V_{h}$ denote an arbitrary finite-dimensional subspace of $H_{0}^{1}\left(I\right)$. We begin by considering the problem given by (Equation 11) with $\phi_{\infty }=0$. We discretize this problem in space using the following FE scheme: For each t∈[0,T], find $\vartheta_{h}\left(t\right)\in V_{h}$ such that $\vartheta_{h}^{0}=\phi_{h}^{0}$ and(12)$$\int_{-L}^{L}\psi_{h}d_{t}\vartheta_{h}\left(t\right)dx+A\left(t,\vartheta_{h},\psi_{h}\right)=0\;\forall \;\psi_{h}\in V_{h},$$
where the bilinear form $A:t\mapsto H_{0}^{1}\left(I\right)\times H_{0}^{1}\left(I\right)$ is defined by(13)$$A\left(t,\vartheta_{h},\psi_{h}\right):=D\int_{-L}^{L}\vartheta_{h}^{\prime }\left(t\right)\psi_{h}^{\prime }dx,$$
and $\phi_{h}^{0}$ is the $L^{2}$-projection of $\phi_{0}$ into $V_{h}$, such that $\phi_{h}^{0}\in V_{h}$ satisfies, for all $\psi_{h}\in V_{h}$,(14)$$\int_{-L}^{L}\phi_{h}^{0}\psi_{h}dx=\int_{-L}^{L}\phi_{0}\psi_{h}dx.$$

To discretize in time, we partition the interval [0,T] as$$0=t^{0}<t^{1}<\cdots <t^{N+1}=T,$$
with time step denoted by $\Delta t^{n}:=t^{n+1}-t^{n}$ for n∈{0,1,…,N}. Furthermore, we denote a function ζ(t) at time level $t=t^{n}$ by $\zeta^{n}$.

For the implicit Euler method in time, we use$${\left(d_{t}\vartheta_{h}\right)}^{n+1}\approx \frac{\vartheta_{h}^{n+1}-\vartheta_{h}^{n}}{\Delta t^{n}}.$$
Inserting this expression into (Equation 12) at time $t^{n+1}$ yields the following fully discrete formulation of (Equation 11) with homogeneous boundary conditions: For each n∈{0,1,…,N}, find $\vartheta_{h}^{n+1}\in V_{h}$, such that(15)$$B\left(\vartheta_{h}^{n+1},\psi_{h}\right)+\Delta t^{n}A\left(\vartheta_{h}^{n+1},\psi_{h}\right)=B\left(\vartheta_{h}^{n},\psi_{h}\right)\;\forall \;\psi_{h}\in V_{h},$$
where, for easy of notation, we write $A(\vartheta_{h}^{n+1},\psi_{h})$ instead of $A(t^{n+1},\vartheta_{h}^{n+1},\psi_{h})$ and(16)$$B\left(\vartheta_{h},\psi_{h}\right):=\int_{-L}^{L}\phi_{h}\psi_{h}dx\;\forall \;\vartheta_{h},\psi_{h}\in V_{h}.$$

We will show that problem (Equation 15) reduces to a system of linear equations. To achieve this, we assume that $M=\dim\;V_{h}<\infty$ and let $\left\{e_{1},\ldots ,e_{M}\right\}$ be a basis of $V_{h}$. Then, for each n∈{0,1,…,N}, there exist $\alpha_{1}^{n+1},\ldots ,\alpha_{M}^{n+1}\in R$ such that(17)$$\vartheta_{h}^{n+1}=\sum_{j=1}^{M}\alpha_{j}^{n+1}e_{j}.$$

We therefore write (Equation 15) as follows: For each n∈{0,1,…,N}, find $\alpha_{1}^{n+1},\ldots ,\alpha_{M}^{n+1}\in R$ such that(18)$$\sum_{j=1}^{M}\alpha_{j}^{n+1}\{B\left(e_{j},e_{i}\right)+\Delta t^{n}A\left(e_{j},e_{i}\right)\}=\sum_{j=1}^{M}\alpha_{j}^{n}B\left(e_{j},e_{i}\right)\;\forall \;i=1,\ldots ,M.$$

In this way, if we set $a_{ij}:=A\left(e_{j},e_{i}\right)$ and $b_{ij}:=B\left(e_{j},e_{i}\right)$, so that$$\alpha^{n+1}:=\left(\alpha_{j}^{n+1}\right)\in R^{M},\;A:=\left(a_{ij}\right)\in R^{M\times M},\;B:=\left(b_{ij}\right)\in R^{M\times M},$$
the matrix form of (Equation 18) reads the following: For each n∈{0,1,…,N}, find $\alpha^{n+1}\in R^{M}$, which satisfies(19)$$\left(B+\Delta t^{n}A\right)\alpha^{n+1}=B\alpha^{n},$$
where the iteration is initialized with $\alpha^{0}\in R^{M}$ obtained from (Equation 14). The following result establishes the unique solvability of the linear system (Equation 19).

**Theorem** **1.**
*The matrix $B+\Delta t^{n}A$ is symmetric and positive definite, and therefore invertible.*


**Proof.** The symmetry property follows directly from the definition of the bilinear forms *A* and *B*. Next, given $\beta :=\left(\beta_{j}\right)\in R^{M}$, we set(20)$$\psi_{h}=\sum_{j=1}^{M}\beta_{j}e_{j}.$$Proceeding analogously to ([36], Chapter 4), that is, using the $H_{0}^{1}\left(I\right)$-ellipticity of the form *A* with ellipticity constant C>0, depending on the diffusion coefficient *D* and the constant from Poincaré’s inequality (cf. (Equation 5)), we obtain$$\begin{array}{cc} \beta^{T}\left(B+\Delta t^{n}A\right)\beta \; & =\sum_{i,j=1}^{M}\left(b_{ij}+\Delta t^{n}a_{ij}\right)\beta_{i}\beta_{j}=B\left(\psi_{h},\psi_{h}\right)+\Delta t^{n}A\left(\psi_{h},\psi_{h}\right)\\ \; & \geq ∥\psi_{h}{∥}_{0,I}^{2}+\Delta t^{n}C∥\psi_{h}{∥}_{1,I}^{2}\geq \Delta t^{n}C{∥\psi_{h}∥}_{1,I}^{2}. \end{array}$$Since $\beta^{T}\left(B+\Delta t^{n}A\right)\beta >0$ for all β values that are different from the null vector, the result follows.    □

#### 2.3.2. Specific FE Subspace

This section specifies the matrix structure of the linear system (Equation 19) for a particular choice of $V_{h}$. Let ${\left\{x_{m}\right\}}_{0\leq m\leq M+1}$ be a uniform partition of the interval $\bar{I}=\left[-L,L\right]$, with the meshsize being denoted by h>0. We set(21)$$V_{h}=\{\psi_{h}\in C\left(\bar{I}\right):\;{\psi_{h}|}_{\left[x_{j-1},x_{j}\right]}\in P_{1}\left(\left[x_{j-1},x_{j}\right]\right)\;\forall \;j=1,\cdots ,M+2\}\cap H_{0}^{1}\left(I\right),$$
where $P_{1}\left(S\right)$ denotes the space of polynomials of degree ≤1 defined over an interval *S*.

The following definition can be found in ([37], Section 1.1.2).

**Definition** **1.**
*For each i∈{1,…,M}, the hat functions $e_{i}\in V_{h}$ are defined as*

$$e_{i}\left(x\right)=\left\{\begin{array}{cc} \frac{x-x_{i-1}}{h} & \;x\in [x_{i-1},x_{i}],\\ \frac{x_{i+1}-x}{h} & \;x\in [x_{i},x_{i+1}],\\ 0 & \;x\notin [x_{i-1},x_{i+1}]. \end{array}\right.$$



An example of a hat function is shown in Figure 2. It follows that $e_{i}\left(x_{j}\right)=\delta_{ij}$, where $\delta_{ij}$ is the Kronecker delta function. Moreover, $\left\{e_{1},\ldots ,e_{M}\right\}$ is a basis for $V_{h}$; hence, a function $\psi_{h}\in V_{h}$ is uniquely determined by the values ${\left\{\psi_{h}\left(x_{i}\right)\right\}}_{1\leq i\leq M}$, namely$$\psi_{h}=\sum_{i=1}^{M}\psi_{h}\left(x_{i}\right)e_{i}.$$

Note that $a_{ij}=b_{ij}=0$ when |i−j|≥2. Furthermore, we obtain, after some algebraic manipulations,$$a_{ij}=\left\{\begin{array}{cc} \frac{2D}{h} & \;\mathrm{if}\;\;j=i,\\ -\frac{D}{h} & \;\mathrm{if}\;\;|j-i|=1. \end{array}\right.$$

Similarly,$$b_{ij}=\left\{\begin{array}{cc} \frac{2h}{3} & \;\mathrm{if}\;\;j=i,\\ \frac{h}{6} & \;\mathrm{if}\;\;|j-i|=1. \end{array}\right.$$

Consequently, the global matrix of (Equation 19) is tridiagonal, which is highly desirable in situations where the dimension of $V_{h}$ is very large, as this allows for a reduction in the number of *flop* required to solve the system. In particular, a tridiagonal system can be solved using the Thomas algorithm, which requires significantly fewer *flop* than the method that directly computes the inverse of the matrix. For further details, we refer the reader to [38].

#### 2.3.3. Numerical Treatment of Non-Homogeneous Boundary Conditions

Recall from (Equation 9) that the unknowns ϕ and ϑ are related by the decomposition ϕ=ϑ+λ, where λ is defined such that $\lambda =\phi_{\infty }$ at x=±L. This approach allows ϕ to be recovered from ϑ, even in the presence of non-homogeneous boundary conditions. Building on this idea, we approximate ϕ using the decomposition(22)$$\phi_{h}^{n+1}=\vartheta_{h}^{n+1}+\lambda_{h},\;n\in \left\{0,1,\ldots ,N\right\},$$
where $\vartheta_{h}^{n+1}\in V_{h}$ is given by (Equation 17), with $V_{h}$ being defined in (Equation 21), and $\lambda_{h}$ is a polynomial satisfying the boundary conditions $\lambda_{h}\left(-L\right)=\lambda_{h}\left(L\right)=\phi_{\infty }$. Specifically, we define(23)$$\lambda_{h}:=\phi_{\infty }\left(e_{0}+e_{M+1}\right),$$
where $e_{0}$ and $e_{M+1}$ are the hat functions associated with the endpoints of *I*.

Next, using the same notation as in Section 2.3.2, the fully discrete formulation of (Equation 11) reads as follows: for each n∈{0,1,…,N}, find $\vartheta_{h}^{n+1}\in V_{h}$ such that $\vartheta_{h}^{0}=\phi_{h}^{0}$ (cf. (Equation 14)) and (24)$$B\left(\vartheta_{h}^{n+1},\psi_{h}\right)+\Delta t^{n}A\left(\vartheta_{h}^{n+1},\psi_{h}\right)=B\left(\phi_{h}^{n},\psi_{h}\right)-\Delta t^{n}D\int_{-L}^{L}\lambda_{h}^{\prime }\psi_{h}^{\prime }dx\;\forall \;\psi_{h}\in V_{h}.$$

It is clear that (Equation 24) coincides with (Equation 15) when $\phi_{\infty }=0$.

Now let $\mu \in R^{M}$ be the vector whose entries are all zero, except for the first and last ones, which are both set to $\phi_{\infty }$. Then, for each n∈{0,1,…,N}, the discretization (Equation 24) yields(25)$$\left(B+\Delta t^{n}A\right)\alpha^{n+1}=B\alpha^{n}-\Delta t^{n}A\mu .$$

Here, the matrices A and B are defined as in Section 2.3.2, thus ensuring the unique solvability of the linear system (Equation 25), which follows directly from Theorem 1.

We end this section by noting that the discrete concentration with non-homogeneous boundary conditions can be computed from (Equation 22) using the solution of (Equation 25), which provides$$\phi_{h}^{n+1}=\sum_{j=0}^{M+1}\alpha_{j}^{n+1}e_{j},\;n\in \left\{0,\cdots ,N\right\},$$
with $\phi_{h}^{n+1}\left(-L\right)=\phi_{h}^{n+1}\left(L\right)=\phi_{\infty }$, as required.

### 2.4. Variable Diffusion Coefficient

This section presents an FE method to solve the non-linear equation in (Equation 1), assuming $\phi_{\infty }$ is time-dependent for greater generality. Although this model is more challenging to analyze with FE methods due to the concentration dependence of *D*, it still relies on the previous results.

The weak form of (Equation 1) resembles (Equation 11), with the form *A* now replaced by$$A\left(\phi ;\vartheta ,\psi \right):=\int_{-L}^{L}D\left(\phi \left(t\right)\right)\vartheta^{\prime }\left(t\right)\psi^{\prime }dx\;\forall \;\psi \in H_{0}^{1}\left(I\right).$$

Note here that ϑ(t):=ϕ(t)−λ(t) and $\lambda \left(t\right)\in H_{0}^{1}{\left(I\right)}^{⊥}$ with $\lambda \left(t\right)=\phi_{\infty }\left(t\right)$ at x=±L.

Proceeding analogously to Section 2.3, we discretize the weak formulation of (Equation 1) using an FE method in space and an implicit Euler method in time. To handle the non-linearities, we employ a Picard-type iteration. Specifically, we consider the following fully discrete scheme: for each n∈{0,1,…,N} and given $\phi_{h}^{n}$, find $\vartheta_{h}^{n+1}\in V_{h}$ such that $\vartheta_{h}^{0}=\phi_{h}^{0}$ and(26)$$\begin{array}{cc} \; & B\left(\vartheta_{h}^{n+1},\psi_{h}\right)+\Delta t^{n}A\left(\phi_{h}^{n};\vartheta_{h}^{n+1},\psi_{h}\right)\\ \; & \;=B\left(\vartheta_{h}^{n},\psi_{h}\right)-\int_{-L}^{L}\lambda_{h}^{n+1}\psi_{h}dx-\Delta t^{n}\int_{-L}^{L}D\left(\phi_{h}^{n}\right){\left(\lambda_{h}^{n+1}\right)}^{\prime }\psi_{h}^{\prime }dx\;\forall \;\psi_{h}\in V_{h}, \end{array}$$
where $\phi_{h}^{0}$ is given by (Equation 14) and *B* by (Equation 16). Furthermore, the last two integrals in (Equation 26) arise from the treatment of the time-dependent boundary condition. In fact, proceeding as in Section 2.3.3, we obtain(27)$$\phi_{h}^{n+1}=\vartheta_{h}^{n+1}+\lambda_{h}^{n+1},\;\mathrm{with}\;\lambda_{h}^{n+1}:=\phi_{\infty }^{n+1}\left(e_{0}+e_{M+1}\right).$$

Now, inspired by [32], we define(28)$$D\left(\phi \right):=D_{0}{\left(1+\delta \phi \right)}^{k},$$
where k∈N, δ∈R, and $D_{0}>0$ are experimentally determined constants. For the sake of simplicity, we assume that *k* is either 1 or 2. However, the analysis in this section can be easily extended to cases where k>2. We furthermore assume that *D* is bounded: There exists a constant $D_{*}>0$ such that, for all ψ∈[0,1],(29)$$D\left(\psi \right)\geq D_{*}.$$

**Remark** **1.**
*As discussed in [39], a general form for the diffusion coefficient is $D\left(\phi \right)=D_{0}F\left(\phi \right)$, where $D_{0}=D\left(0\right)$ is a positive constant. The choice in (Equation 28), with $F\left(\phi \right)={\left(1+\delta \phi \right)}^{k}$, represents a specific instance of this general form. While other forms of F, such as exponential functions, have been explored in the literature (see, e.g., [40]), we restrict our focus to (Equation 28) for brevity.*


Next, we specify the matrix form of the fully discrete scheme (Equation 26), subjected to the diffusion coefficient (Equation 28). Let $\hat{A}:=\left({\hat{a}}_{ij}\right)\in R^{M\times M}$ denote the matrix given by(30)$${\hat{a}}_{ij}:=A\left(\phi_{h}^{n};e_{j},e_{i}\right)=\int_{-L}^{L}D\left(\phi_{h}^{n}\right)e_{i}^{\prime }e_{j}^{\prime }dx=D_{0}\int_{-L}^{L}({1+\delta \sum_{l=0}^{M+1}\alpha_{l}^{n}e_{l})}^{k}e_{i}^{\prime }e_{j}^{\prime }dx,$$
where $e_{0},e_{1},\ldots ,e_{M+1}$ are the hat functions introduced in Section 2.3.2.

After some algebraic manipulations, we obtain$$\hat{A}=\frac{D_{0}}{h}\left[\begin{array}{ccccccc} r_{1} & q_{1} & 0 & 0 & \cdots & \cdots & 0\\ q_{1} & r_{2} & q_{2} & 0 & \cdots & \cdots & 0\\ 0 & \ddots & \ddots & \ddots & \ddots & \; & 0\\ \vdots & \ddots & \ddots & \ddots & \ddots & \ddots & \vdots \\ 0 & \cdots & 0 & q_{M-3} & r_{M-2} & q_{M-2} & 0\\ 0 & \cdots & \cdots & 0 & q_{M-2} & r_{M-1} & q_{M-1}\\ 0 & \cdots & \cdots & 0 & 0 & q_{M-1} & r_{M} \end{array}\right],$$
with matrix components depending on the value of *k*. Specifically, for k=1,$$\begin{array}{cc} q_{i} & :=-1-\frac{\delta }{2h}\left(\alpha_{i}^{n}+\alpha_{i+1}^{n}\right),\\ r_{i} & :=2+\frac{\delta }{2h}\left(\alpha_{i-1}^{n}+2\alpha_{i}^{n}+\alpha_{i+1}^{n}\right), \end{array}$$
while for k=2,$$\begin{array}{cc} q_{i} & :=-\frac{1}{3}{\left(1+\delta \alpha_{i}\right)}^{2}-\frac{1}{3}\left(1+\delta \alpha_{i}\right)\left(1+\delta \alpha_{i+1}\right)-\frac{1}{3}{\left(1+\delta \alpha_{i+1}\right)}^{2},\\ r_{i} & :=\frac{1}{3}{\left(1+\delta \alpha_{i-1}\right)}^{2}+\frac{2}{3}{\left(1+\delta \alpha_{i}\right)}^{2}+\frac{1}{3}{\left(1+\delta \alpha_{i+1}\right)}^{2}+\frac{1}{3}\left(1+\delta \alpha_{i}\right)\left(2+\delta \alpha_{i-1}+\delta \alpha_{i+1}\right). \end{array}$$

Finally, for each n∈{0,1,⋯,N}, we set $\mu^{n+1}\in R^{M}$ as the vector whose components are all zero, except for the first and last ones, which are set to $\phi_{\infty }^{n+1}$. Then, the matrix form of (Equation 26) reads as follows: for each n∈{0,1,…,N}, find $\alpha^{n+1}:=\left(\alpha_{j}^{n+1}\right)\in R^{M}$ such that(31)$$\left(B+\Delta t^{n}\hat{A}\right)\alpha^{n+1}=B\alpha^{n}-\left(B+\Delta t^{n}\hat{A}\right)\mu^{n+1},$$
where the matrix B is defined as in Section 2.3.2.

**Remark** **2.**
*The unique solvability of the fully discrete scheme (Equation 31) follows from the fact that the diffusion coefficient (Equation 28) satisfies the condition (Equation 29). In fact, the matrix $B+\Delta t^{n}\hat{A}$ is symmetric, and for any vector $\beta \in R^{M}$ given by (Equation 20), we have*

$$\beta^{T}\left(B+\Delta t^{n}\hat{A}\right)\beta \geq \Delta t^{n}C{∥\psi_{h}∥}_{1,I}^{2},$$

*with constant C>0 depending on $D_{*}$ and the constant from Poincaré’s inequality. It then follows that the matrix $B+\Delta t^{n}\hat{A}$ is positive definite, and therefore invertible.*


## 3. Results

### 3.1. Drug Release Experiments

Samples of NC/nPSi composites (0.5 × 0.5 ${\mathrm{cm}}^{2}$) with varying concentrations of microparticulate nPSi were synthesized following the protocol outlined by K. Garrido-Miranda et al. [12]. Each sample had different thicknesses based on the percentage of nPSi (m/m): NC (control, 0.0%) = 6.5 ± 1 μm, NC/nPSi-0.1% = 10.5 ± 2 μm, NC/nPSi-0.5% = 12.7 ± 3 μm, and NC/nPSi-1.0% = 29.5 ± 4 μm (Figure 3). Subsequently, the samples were loaded with a concentrated MB solution (0.001 M, pH 7.0) for 15 min at 100 rpm. They were then rinsed with distilled water and dried at room temperature. Release profiles were conducted in vials filled with 3 ml of saline phosphate-buffered solution (PBS, pH 7 and 37 °C) at 100 rpm on a horizontal shaker (INB-2005 LN, Biotek, Winooski, VT, USA). The concentration of MB in the fluid was measured at specific time intervals using UV-Vis spectrophotometry (UV-1800 Shimadzu, Kyoto, Japan) at a wavelength of 671 nm [7]. The release profiles were obtained as the mean of triplicate experiments (see the Appendix A section).

### 3.2. Model Prediction

In this section, we compare the drug release experiments with the numerical results obtained using the FE method described in Section 2.4. All simulations were implemented using Octave 8.4.0 [33].

In what follows, the diffusion coefficient *D* is defined as in (Equation 28), with the constant $D_{0}$ determined using the Nelder–Mead optimization method [41], which minimizes the error$$e_{ana}:=\sqrt{\sum_{n=0}^{N+1}{\left|u^{n}-w^{n}\left(D_{0}\right)\right|}^{2}},$$
where $u^{n}$ represents the experimental release profile at time $t^{n}$, and $w^{n}\left(D_{0}\right)$ denotes the corresponding value obtained from the analytical release profile in (Equation 3), computed as$$w^{n}\left(D_{0}\right)=1-\frac{8}{\pi^{2}}\sum_{k=0}^{p}\frac{1}{{\left(2k+1\right)}^{2}}\exp[-\frac{D_{0}{\left(2k+1\right)}^{2}\pi^{2}t^{n}}{4L^{2}}].$$

Note here that *p* is a user-defined integer, which we set to p=100. The computed values of $D_{0}$ for the four samples detailed in Section 3.1 are listed in Table 1. Once the parameter $D_{0}$ is determined, we proceed to complete the definition of the diffusion coefficient *D* by selecting an appropriate value for the scalar δ, as will be discussed in more detail below. With *D* being fully defined, the release profile $M_{t}/M_{\infty }$ is approximated by solving Fick’s second law using the FE method. Specifically, the solution to the system (Equation 31), as obtained from Algorithm 1, provides the following approximation: (32)$${\frac{M_{t}}{M_{\infty }}|}_{t=t^{n+1}}\approx \frac{\int_{-L}^{L}\left(\phi_{h}^{n+1}-\phi_{0}\right)dx}{\int_{-L}^{L}\left(\phi_{\infty }^{n+1}-\phi_{0}\right)dx}.$$

Unless otherwise specified, we choose $\phi_{0}=1$ and $\phi_{\infty }^{n+1}=0$ for all n∈{0,1,…,N+1}.
**Algorithm 1:** FE solution.     **Input**: *N*, *M*, δ, $D_{\mathrm{ref}}$, $\phi_{0}$, $\phi_{\infty }$, *L*     **Output**: $\phi_{h}^{0},\phi_{h}^{1},\ldots ,\phi_{h}^{N+1}$1:h←2L/(M+1)2:$\phi_{h}^{0}\leftarrow$$L^{2}$-projection of $\phi_{0}$ into $V_{h}$ (cf. (Equation 14))3:**for **n=0,1,…,N** do**4:    $\Delta t^{n}\leftarrow t^{n+1}-t^{n}$5:    $\alpha_{0}^{n+1}\leftarrow \phi_{\infty }^{n+1}$6:    $\alpha_{M+1}^{n+1}\leftarrow \phi_{\infty }^{n+1}$7:    $\alpha_{1}^{n+1},\ldots ,\alpha_{M}^{n+1}\leftarrow$ solution to the problem (Equation 31)8:    $\phi_{h}^{n+1}\leftarrow \sum_{j=0}^{M+1}\alpha_{j}^{n+1}e_{j}$9:**end for**

Selecting the optimal values for δ and *k* is essential for accurately defining the diffusion coefficient *D*. To determine these values, we focused on the samples in Table 1. For each sample, *k* was set to either 1 or 2, the meshsize *h* was computed as 2L/(M+1) with M=31, and δ was chosen from the interval [−1,1]. The goal was to minimize the numerical error(33)$$e_{num}:=\sqrt{\sum_{n=0}^{N+1}{\left|u^{n}-v^{n}\left(\delta \right)\right|}^{2}},$$
where the experimental release profile at time $t^{n}$ is denoted by $u^{n}$, while $v^{n}\left(\delta \right)$ represents the corresponding approximation from (Equation 32). Values of δ outside the interval [−1,1] led to larger errors and were therefore discarded. In this context, the computed errors for k=2 are shown in Table 2, indicating that the optimal value of δ is 0.3 for the first three samples and 0.2 for the last one. Thus, condition (Equation 29) is satisfied with $D_{*}=D_{0}$, ensuring the unique solvability of the fully discrete scheme (Equation 26), as stated in Remark 2. Similar results for k=1 were obtained and are omitted for brevity.

Figure 4 shows the experimental release data, their numerical approximations using the FE method, and the analytical solution from (Equation 3). The FE solution provided the best fit for the experimental data in the first three samples. For the last sample, where the concentration of nPSi was higher, the analytical release profile was 20.55% more accurate, as indicated by the errors in Table 2. It follows that *D* strongly depends on the concentration when the percentage of nPSi is below 1.0%. Additionally, the simulations in Figure 5 reveal that the diffusion rate decreases at higher nPSi percentages, as reflected by the slower reduction in concentration across the thickness, particularly in the last sample. These results align with the experimental findings of K. Garrido-Miranda et al. [12], where incorporating nPSi into the material enhances control over the release of MB.

## 4. Discussion

The FE method presented in Section 2.4 provides a general framework for accurately describing the drug delivery system under consideration. In fact, the variable diffusion coefficient defined in (Equation 28) showed strong agreement with data from K. Garrido-Miranda et al. [12], particularly for samples with nPSi percentages below 1.0%. A weaker dependency was observed at an nPSi concentration of exactly 1.0%. This may be linked to the increased thickness associated with higher nPSi levels, which was presumed negligible in both the numerical method and the analytical solution.

In our simulations, we assumed constant initial and boundary conditions. However, a more realistic assumption would involve time-dependent boundary data, which may occur when the solution is not well stirred [42]. Notably, the fully discrete scheme (Equation 26) remains effective in simulating this scenario, as illustrated in Figure 6 for a single sample, where the boundary data were derived from the empirical relation $\phi_{\infty }\left(t\right)=1-M_{t}/M_{\infty }$. These results were contrasted with the analytical solution from (Equation 3), showing that the numerical solution with time-dependent $\phi_{\infty }$ performs better when $M_{t}/M_{\infty }<0.85$. This conclusion is further supported by the error computed from (Equation 33) using δ=0.45 and k=2, yielding a value of 0.149, which represents a 14.77% improvement compared to the corresponding error reported in Table 2 for the analytical solution with constant boundary conditions.

Building on these results, a more comprehensive approach to designing drug delivery systems that undergo significant geometric changes and time-dependent boundary conditions should incorporate space–time FE methods (see, e.g., [43,44]). Future work will explore this approach by incorporating additional parameters into the general form of diffusion coefficient presented in Remark 1, with the aim of capturing effects such as evaporation within the composite.

## 5. Conclusions

This study employed an FE method to model diffusion and controlled drug release from NC/nPSi composites based on Fick’s second law with variable diffusivity. The FE simulations, supported by experimental validation, demonstrated that increasing the nPSi concentration in the NC matrix enhances control over the release MB. This finding is particularly relevant for pharmaceutical applications, where controlled drug delivery is critical to minimize adverse effects on tissues. Furthermore, this study highlights the importance of considering geometric changes in the composite matrix, especially at high nPSi concentrations, which significantly influence drug release behavior. Addressing these complexities is a priority for future work to extend the applicability of the proposed model to a wider range of material configurations and drug delivery scenarios. The results not only validate the effectiveness of the FE method for modeling diffusion in composite materials but also provide insights into optimizing the design of NC/nPSi composites for controlled drug release. This work sets a foundation for further research into more complex geometries and dynamic environmental conditions, reinforcing its potential contributions to the development of advanced drug delivery systems.

## Figures and Tables

**Figure 1 pharmaceutics-17-00120-f001:**
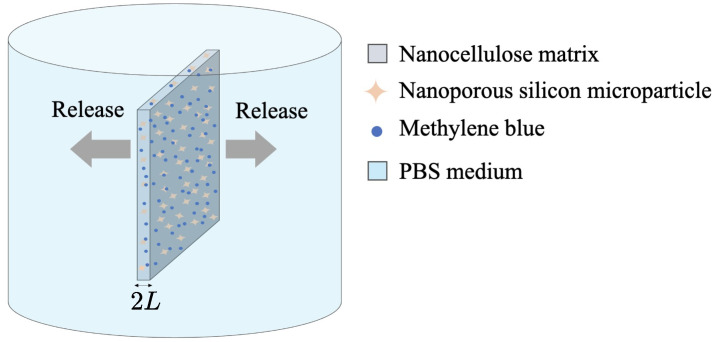
Scheme of the drug delivery system.

**Figure 2 pharmaceutics-17-00120-f002:**
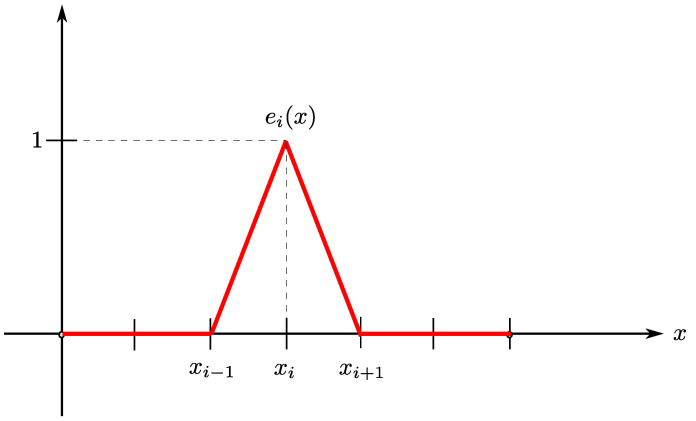
Example of hat function.

**Figure 3 pharmaceutics-17-00120-f003:**
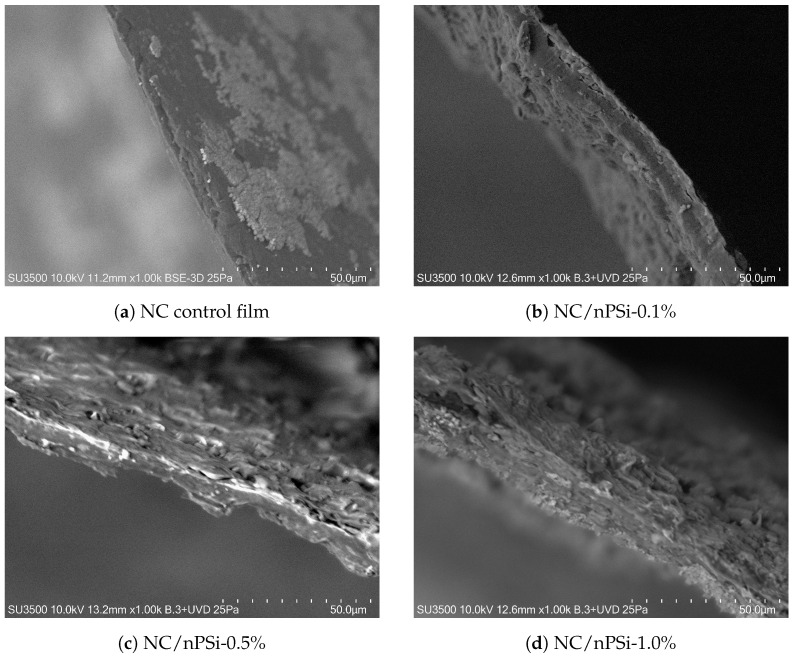
SEM images of samples.

**Figure 4 pharmaceutics-17-00120-f004:**
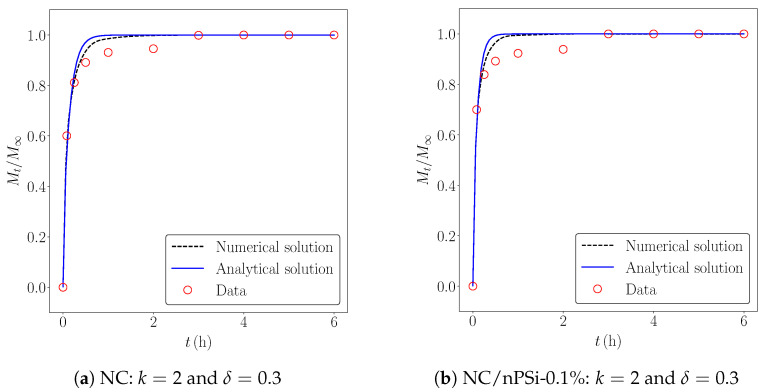
Numerical vs. analytical drug release profiles.

**Figure 5 pharmaceutics-17-00120-f005:**
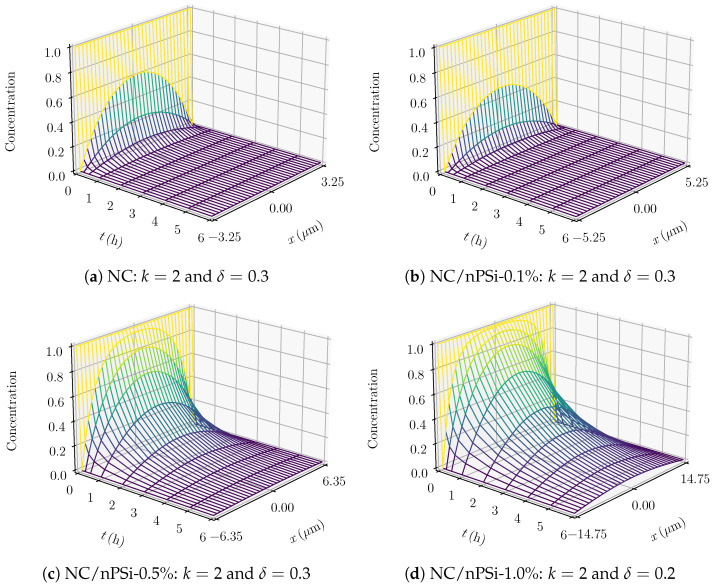
Approximate concentration.

**Figure 6 pharmaceutics-17-00120-f006:**
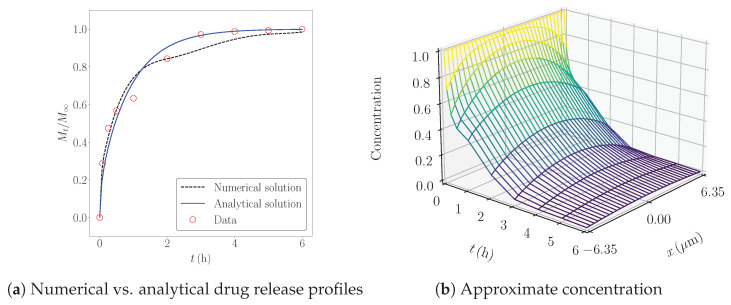
NC/nPSi-0.5%: k=2 and δ=0.45.

**Table 1 pharmaceutics-17-00120-t001:** Computed values of $D_{0}$.

Sample	$D_{0}$ (${\mu m}^{2}$/h)
NC	28.648
NC/nPSi-0.1%	107.640
NC/nPSi-0.5%	17.413
NC/nPSi-1.0%	51.523

**Table 2 pharmaceutics-17-00120-t002:** Release profile of errors associated with the variable diffusion coefficient *D* for k=2.

NC							
δ	−1	−0.3	−0.2	0	0.2	0.3	1
$e_{num}$	1.636	0.320	0.262	0.166	0.104	0.092	0.200
$e_{ana}$	–	–	–	0.140	–	–	–
NC/nPSi-0.1%							
δ	−1	−0.3	−0.2	0	0.2	0.3	1
$e_{num}$	1.573	0.333	0.279	0.194	0.145	0.135	0.193
$e_{ana}$	–	–	–	0.178	–	–	–
NC/nPSi-0.5%							
δ	−1	−0.3	−0.2	0	0.2	0.3	1
$e_{num}$	1.344	0.314	0.264	0.185	0.141	0.134	0.263
$e_{ana}$	–	–	–	0.171	–	–	–
NC/nPSi-1.0%							
δ	−1	−0.3	−0.2	0	0.2	0.3	1
$e_{num}$	1.219	0.306	0.257	0.191	0.176	0.185	0.351
$e_{ana}$	–	–	–	0.146	–	–	–

## Data Availability

The data presented in this study are available in the Appendix A.

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
