# Peer review of "A Finite Element Method for Modeling Diffusion and Drug Release from Nanocellulose/Nanoporous Silicon Composites"

_pharmaceutics, 2025, doi:10.3390/pharmaceutics17010120_

Round 1
Reviewer 1 Report
Comments and Suggestions for Authors
Advantages:
1. The article presents a novel concentration-dependent diffusion coefficient model, which more deeply reflects the kinetics of drug release in composite materials.
2. The research involves nanocellulose and nanoporous silicon composite materials, both of which have potential for exploration in drug delivery.
3. Wide applicability of the method:
This study provides a theoretical basis and research foundation for the release mechanism of special drugs, and offers a mathematical modeling framework for the study of other more complex drug release systems.
Disadvantages:
Firstly, the data in the research results is slightly lacking. It is hoped that more substantive factual evidence can be provided to support the theories proposed in the article. For example, some experiments can be added as supplementary evidence to the case study section. Secondly, the scale bars in the SEM micrographs should be marked on the images themselves. The (c) and (d) images are not clear. In summary, it is suggested to add some materials science experiments to enhance the persuasiveness of the case study section.
Author Response
The response letter to the Reviewers is attached.

Reviewer 2 Report
Comments and Suggestions for Authors
The manuscript “A Finite Element Method for Modeling Diffusion and Drug Release from Nanocellulose/Nanoporous Silicon Composites” represents the development of a finite element (FE) method to solve the Fick equations governing the diffusion and controlled release of methylene blue from nanocellulose and nanoporous silica composites. The authors carried out a great deal of calculation work. The theoretical calculation is supported by practical results, which is the advantage of this work. The text is structured, all necessary sections are present. The findings are consistent with the results of the study.
However, after reading some comments arise. In the introduction, the authors say nothing about the dependence of the diffusion coefficient on temperature. Have the authors studied this issue? It seems important to study the change in the diffusion coefficient at 36.5; 37.0; 37.5, since a change in temperature even by 1 degree in a biological environment is a significant aspect. If the authors did not study the temperature dependence, then the influence of temperature on the diffusion process should still be mentioned in the Introduction.
Author Response

(The authors gave the same response as above.)

Reviewer 3 Report
Comments and Suggestions for Authors
Dear Author,
A Finite Element Method for Modeling Diffusion and Drug Release from Nanocellulose/Nanoporous Silicon Composites is an interesting article and topic presented nicely to convey story to audience.I have following comments to improve quality of paper:
1.Abstract should be formatted like Objective, method ,results, and conclusion so easier to follow reader.
2.Proof. See [32, Proposition 8.13]. line 92 remove square typo,
3.Figure 3. SEM images of samples: (a) NC control film, (b) NC/nPSi 0.1%, (c) NC/nPSi 0.5%, and (d) NC/nPSi 1.0%. Scales bar: 10 µm. add erroer bar to fig.
4.Figure 4. and Figure 6. move this figures to results.
Comments on the Quality of English LanguageNA
Author Response

(The authors gave the same response as above.)
